# Prepubertal Periodontitis in a Patient with Combined Classical and Periodontal Ehlers–Danlos Syndrome

**DOI:** 10.3390/biom11020149

**Published:** 2021-01-24

**Authors:** Friedrich Stock, Marcel Hanisch, Sarah Lechner, Saskia Biskup, Axel Bohring, Johannes Zschocke, Ines Kapferer-Seebacher

**Affiliations:** 1Institute of Human Genetics, University Hospital Münster, Vesaliusweg 12, D-48149 Münster, Germany; Friedrich.stock@ukmuenster.de (F.S.); axel.bohring@ukmuenster.de (A.B.); 2Research Unit Rare Diseases with Orofacial, Manifestations (RDOM), Department of Cranio-Maxillofacial Surgery, University Hospital Münster, Albert-Schweitzer-Campus 1, Building W 30, D-48149 Münster, Germany; marcel.hanisch@ukmuenster.de; 3Praxis für Humangenetik Tübingen, Paul-Ehrlich-Straße 23, D-72076 Tübingen, Germany; info@cegat.de (S.L.); info@humangenetik-tuebingen.de (S.B.); 4Institute of Human Genetics, Medical University of Innsbruck, Peter-Mayr-Straße 1, A-6020 Innsbruck, Austria; johannes.zschocke@i-med.ac.at; 5Department of Operative and Restorative Dentistry, Medical University of Innsbruck, Anichstraße 35, A-6020 Innsbruck, Austria

**Keywords:** Ehlers−Danlos syndrome, periodontitis, complement, C1R, type V collagen

## Abstract

We report an extremely rare case of combined classical and periodontal Ehlers−Danlos syndrome (EDS) with early severe periodontitis and a generalized lack of attached gingiva. A German family with classical EDS was investigated by physical and dental evaluation and exome and Sanger sequencing. Due to the specific periodontal phenotype in the affected child, an additional diagnosis of periodontal EDS was suspected. Physical and genetic examination of two affected and three unaffected family members revealed a family diagnosis of classical EDS with a heterozygous mutation in COL5A1 (c.1502del; p.Pro501Leufs*57). Additional to the major clinical criteria for classical EDS—generalized joint hypermobility, hyperelastic skin, and atrophic scarring —the child aged four years presented with generalized alveolar bone loss up to 80%, early loss of two lower incisors, severe gingival recession, and generalized lack of attached gingiva. Due to these clinical findings, an additional diagnosis of periodontal EDS was suspected. Further genetic analysis revealed the novel missense mutation c.658T>G (p.Cys220Gly) in C1R in a heterozygous state. Early severe periodontitis in association with generalized lack of attached gingiva is pathognomonic for periodontal EDS and led to the right clinical and genetic diagnosis in the present case.

## 1. Introduction

The Ehlers–Danlos syndromes (EDSs) comprise a clinically and genetically heterogeneous group of connective tissue disorders characterized by variable combinations of joint hypermobility, skin hyperextensibility, connective tissue fragility, and other manifestations. In 2017 the international EDS consortium published a revised classification system, “The 2017 International Classification of the Ehlers–Danlos Syndromes” [1], which recognizes 13 distinct types of EDS caused by mutations in 19 different genes (the genetic basis of one relatively common type, hypermobile EDS, is unknown). Two additional conditions were described in 2018 and 2020 [1,2,3]. In contrast to the previous EDS classifications [4,5] the new system combines clinical presentation and pathogenic considerations in the delineation of the different EDS types. 

Classical EDS (cEDS) is inherited in an autosomal-dominant pattern and is characterized by the major criteria hyperextensibility of the skin, atrophic scarring, and generalized joint hypermobility [1]. Affected individuals have soft and doughy skin, which can be extended easily and snaps back after release. Due to the fragility of the skin, even relatively minor trauma can lead to bruising and cuts, especially over pressure points (knees, elbows, shins, forehead, and chin). Joint hypermobility affects both large and small joints, depending on age, gender, family, and ethnic background. It usually becomes obvious when a child begins to walk. Dislocations of the shoulder, patella, digits, hip, radius, and clavicle occur frequently and usually resolve spontaneously or are easy to repone by the affected person. Wounds heal slowly and often result in atrophic and papyraceous (“cigarette-paper”-like), sometimes hemosiderotic scars. Minor diagnostic criteria are dermatologic findings like molluscoid pseudotumors, subcutaneous spheroids, and piezogenic papules. Other features include hypotonia with delayed motor development, fatigue, and muscle cramps, hernias, mitral valve prolapse, and aortic root dilatation. According to the 2017 classification, classical EDS is clinically diagnosed when skin hyperextensibility and atrophic scarring is associated with either generalized joint hypermobility or at least three minor criteria [1]. The clinical diagnosis of cEDS can be confirmed by detection of a heterozygous pathogenic mutation in one of the genes encoding type V collagen (*COL5A1* and *COL5A2*) in more than 90% of patients; rarely specific mutations in the genes encoding type I collagen (*COL1A1* and *COL1A2*) are detected.

Periodontal EDS (pEDS) is a widely unknown condition. It is characterized by the major criteria of severe periodontitis of early-onset, lack of attached gingiva, pretibial hemosiderin depositions, and family history of a first-degree relative who meets the clinical criteria [1]. Affected individuals develop extensive gingivitis on even mild biofilm accumulation in their childhood, leading to periodontal tissue destruction in adolescence at the latest and premature loss of teeth [6]. Lack of attached gingiva predisposing for gingival recession is a pathognomonic feature of pEDS and the only clinical finding that is consistently present already in childhood [7]. Easy bruising and unresolving hematomas result in characteristic haemosiderin depositions on the shins. Minor criteria of pEDS include joint hypermobility, skin hyperextensibility and fragility, an increased rate of infections, a Marfanoid facial appearance, acrogeria, and prominent vasculature. The diagnosis pEDS can be suggested when three major criteria and one minor criterion are fulfilled [1]. To confirm the diagnosis of pEDS, a heterozygous gain-of-function mutation in the genes encoding subunits C1r and C1s of the first component of the classical complement pathway (*C1R* and *C1S*) must be detected. pEDS mutations cause constitutive intracellular activation of C1s (and C1r) serine proteases resulting in C4 cleavage and local complement cascade activation [8]. Activation of the complement pathway by host-microbe interactions like immunoglobulins bound to antigens has been shown to promote inflammatory bone loss in periodontitis [9]. Periodontitis patients present significantly higher serum and salivary immunoglobulins against periopathogens compared to healthy controls [10]. Additionally, it was previously shown that C1 proteins of low concentration could significantly enhance the expression of interleukins IL-6, IL-8, and IL-10 [11], which is in line with the finding that periodontitis patients have significantly higher salivary IL-6 levels than healthy subjects [12].

The current classification of periodontal and peri-implant diseases states that generalized severe periodontitis is a manifestation of periodontal EDS but may also occur in vascular EDS (characterized by a high risk of arterial and internal organ ruptures) and to a lesser extent in classical EDS [13]. Here we report a unique case of a child with combined classical and periodontal EDS, diagnosed through careful clinical examination, including dental and periodontal investigations. The rationale of the present case study was to highlight the need for astute clinical characterization of individuals with rare diseases in order to recognize the presence of two independent monogenic conditions. We also aimed to illustrate that specific periodontal findings are characteristic for periodontal EDS but not for other EDS subtypes.

## 2. Case Report

### 2.1. Medical History and Physical Examination of the Index Patient

The index patient is an almost five-year-old girl, the second child of non-consanguineous parents of German ancestry. She was born via cesarean section after 36 weeks of gestation. At birth, she weighed 2340 g (10th–15th centile), body length was measured 44.5 cm (ca. 4th centile), head circumference 31 cm (ca. 4th centile). 

Due to bilateral club feet, she received intensive physiotherapy for eight months. At the age of four months, bilateral pes adductus, central coordination disturbance, and asymmetry of posture and tonus were noted. Those signs were retrospectively attributed to joint hypermobility. Hypermobility of the ankles was already noticeable as a baby, as she often twisted her ankle. Due to recurring luxation of the patella and chronic joint pain at the age of four years, the girl was provided with several auxiliary devices and treated with orthoses and bandages for knees and ankles.

From the end of her first year of life, she developed bluish discolorations on her shins and forearms. Falls or minor hits resulted in extensive and persistent hematomas (Figure 1). Her skin was very soft and hyperelastic; there were multiple subcutaneous papules (especially on both shins and her left forearm) and few atrophic scars (“cigarette paper scars”). 

At the age of four years eight months, the proband had a short stature, her height constantly at the second centile, and her BMI were in a regular range. Clinical examination revealed generalized muscular hypotonia, hypermobility of all joints (Beighton score 8), severe flat feet (pes planovalgus), a lumbar lordosis hollow back, a left-convex scoliosis (19.1° Cobb) leg length difference (+4 mm right), and craniomandibular dysfunction. Based on these findings, a diagnosis of classical EDS was suspected.

### 2.2. Family History

Prior to the diagnosis of classical EDS at four years of age in the index patient, the mother and father regarded themselves as healthy. On specific inquiry, the proband’s mother reported clinical features highly indicative of classical EDS beginning in her childhood, including joint hypermobility, umbilical hernia, easy bruising (in particular recurring hematomas on her knees), and poor wound healing. She denied any significant dental or eye problems but reported recurrent airway infections. On careful clinical examination, she had hyperelastic and soft skin, multiple subcutaneous papules, several atrophic scars, piezogenic papules on her heels, and general joint hypermobility with a Beighton score of 5. Periodontal investigations displayed a healthy periodontium with community periodontal index of treatment needs (CPITN) grade 0 and no radiological bone loss. Intraorally, she had a nice band of attached gingiva (Figure 1 and Figure 2).

The proband’s two siblings (sister +3, brother 4 years) and the mother’s sister [III:3] did not display any signs of EDS. The maternal grandfather [II:2], 59 years old, was reported to have a similar phenotype as the mother (see Table 1). Unfortunately, he declined clinical examination and genetic testing. No relevant clinical information was available for his deceased parents. 

### 2.3. Oral Examination

Due to premature loss of two lower incisors at four years of age, the proband was referred to the Clinic for Oral and Maxillofacial Surgery of the University Hospital Münster for a special consultation on “Rare Diseases with Oral Manifestations”. The mother reported that the teeth had fallen out without root resorption. Clinical examination showed a primary dentition with missing teeth 71 and 81. In all four quadrants, CPITN was grade I revealing gingival inflammation and probing pocket depths ≤3 mm. Furcation involvement grade III was diagnosed for deciduous molars 54, 64, 74, and 84. Lack of attached gingiva was diagnosed, and severe gingival recession up to 6 mm was found for all primary teeth except the lower second molars (Figure 4B). Radiologically, generalized alveolar bone loss of up to 80% was diagnosed (Figure 4A). Due to early severe periodontal destruction and lack of attached gingiva periodontal EDS was suspected. No periodontal destruction was observed at the mother (Figure 3 and Figure 4).

### 2.4. Molecular Analysis and Findings

Massive-parallel sequence analysis of a panel of 44 connective tissue-related genes using in-solution hybridization enrichment (Agilent, Santa Clara, CA, USA) and the Illumina HiSeq2500system (Illumina, San Diego, CA, USA) was performed in the index patient. This analysis revealed a heterozygous single nucleotide deletion c.1502del in exon 12 of the *COL5A1* gene; which is predicted to generate a frameshift with a stop codon after 57 amino acids, denoted p.Pro501Leufs*57. It most probably leads to nonsense mediated mRNA-decay, resulting in haploinsufficiency of the alpha-1 chain of collagen V. Sanger sequence analysis in the parents that the mother, but not the father, also carried the *COL5A1* mutation. No other family member was available for genetic testing. Based on clinical and genetic findings, the first clinical diagnosis was classical EDS. 

Six months later, due to periodontal involvement and lack of attached gingiva, it was hypothesized that the proband could also be affected by periodontal EDS. Molecular analysis of the genes *C1R* and *C1S* was performed by PCR amplification and Sanger sequencing, using standard methods. This analysis revealed the novel *C1R* missense variant c.658T>G in a heterozygous state. This variant is predicted to replace a highly conserved cysteine residue with glycine in the “R” subunit of the C1r protein, denoted p.(Cys220Gly). It is not listed in population databases such as gnomAD and is not listed in the disease variant databases ClinVar or HGMD. The analysis software MutationTaster, fathmm, Mutation Assessor, SIFT, fathmm-MKL coding, LRT, and PROVEAN consistently consider this variant as pathogenic. By Sanger sequencing, this mutation was excluded in the parents and thus confirmed to be de novo in our index. On these premises, as well as on pathophysiological considerations, we classify the variant as likely pathogenic, confirming the clinical diagnosis of periodontal EDS.

## 3. Discussion

Here we report a unique case of classical EDS with a frameshift mutation in *COL5A1* in a German family, combined with periodontal EDS caused by a de novo *C1R* in an almost five-year-old girl. This is the second case of two concomitant types of EDS in one person, the first being a family with periodontal and vascular EDS published in 2018 [14]. Both reports strengthen the authors’ observation that early severe periodontal destruction in association with lack of attached gingiva is a specific finding of periodontal EDS but not of other EDS types. 

Totally one hundred eight individuals with molecularly confirmed periodontal EDS have been reported until now, with early severe periodontitis diagnosed in 99% of adult individuals [6,7,14,15,16]. Initiation of periodontal destruction of the permanent dentition is suspected in the early teens [6,17]. A recent clinical study on twelve children aged four to 13 years with molecularly confirmed pEDS showed that the only consistent finding at that age was a generalized lack of attached gingiva [7]. In this cohort, only the oldest proband at age 13 years presented with periodontal destruction (localized clinical attachment loss of up to 6 mm) [7]. Periodontal destruction of the primary dentition has not been reported up to now, although the premature loss of some deciduous teeth has been retrospectively reported in single affected individuals.

In contrast to these data, severe (inflammatory) periodontitis has never been reported in any case of molecularly confirmed classical EDS up to now [17]. The usually cited reference for periodontitis in classical EDS is the paper by Pope et al. (1992), where the authors describe three patients with defective dentinogenesis affecting the mandibular incisors [18]. Some similar cases were described [19,20]. In one of these individuals, two mandibular incisors with hypoplastic roots presented with severe periodontal breakdown [18]. Based on this report and the underlying pathogenesis, i.e., a connective tissue disease, periodontal breakdown with classical EDS is very rare and should be rather classified as (non-inflammatory) tooth-loss in the category “systemic diseases affecting the periodontal supporting tissues“ than in the category “periodontitis as a manifestation of systematic diseases“ of the current classification of periodontal and peri-implant diseases. 

For individuals affected by EDS, it is quite important to distinguish between these clinical nuances. Classical and periodontal EDS are two specific entities with different treatment needs, especially in early childhood. With classical EDS, only symptomatic treatment of the skin and of joint-related symptoms is possible. Physiotherapy may slow down joint degeneration and reduce pain. In severe cases, joints can be stabilized surgically or by specific orthoses. Most patients need pain therapy, either permanently or occasionally. In the case of slowly healing wounds or scarring (especially in delicate locations), plastic surgery may be needed. With periodontal EDS, treatment in childhood mainly focuses on dental prophylaxis consisting of strict dental hygiene to stop biofilm-associated periodontal hyperinflammation and subsequent bone and tooth loss. In general, dentists play an important role in therapy and early diagnoses of underlying diseases based on periodontally relevant manifestations [21].

Many other inflammatory conditions like mast cell activation disorder, ankylosing spondylitis, rheumatoid or inflammatory arthritis may occur with EDS, particularly together with hypermobile EDS [22]. Although an association has been demonstrated between some EDS types and these inflammatory conditions, there is not yet enough scientific evidence to prove that one issue causes the other.

## 4. Conclusions

Our case highlights the need to have an open mind with regard to the possibility of two (or more) independent monogenic conditions in the same individual. This should be considered if clinical features in a genetically defined disease do not fully fit the known spectrum of phenotypes. Severe periodontitis with early-onset is the hallmark of pEDS, but there is no evidence that it is also part of the clinical phenotype of other forms of EDS. Clinical cohort studies on different types of EDS are needed to define specific dental and periodontal manifestations.

## Figures and Tables

**Figure 1 biomolecules-11-00149-f001:**
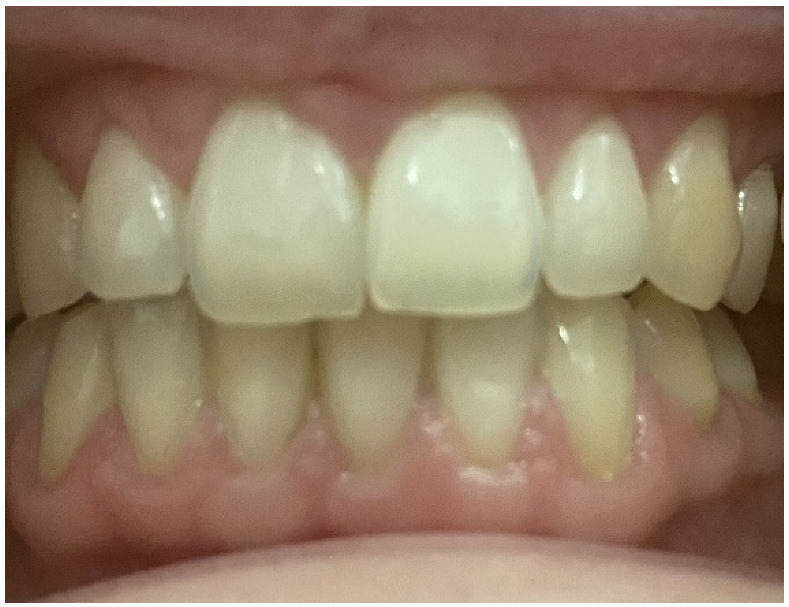
Clinical dental investigation of the mother.

**Figure 2 biomolecules-11-00149-f002:**
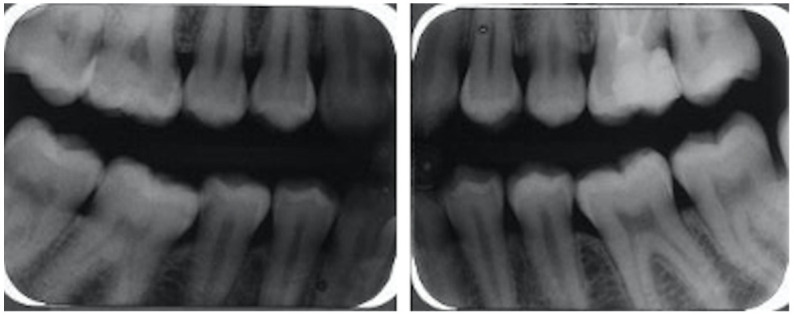
Radiological investigation of the mother demonstrating no bone loss.

**Figure 3 biomolecules-11-00149-f003:**
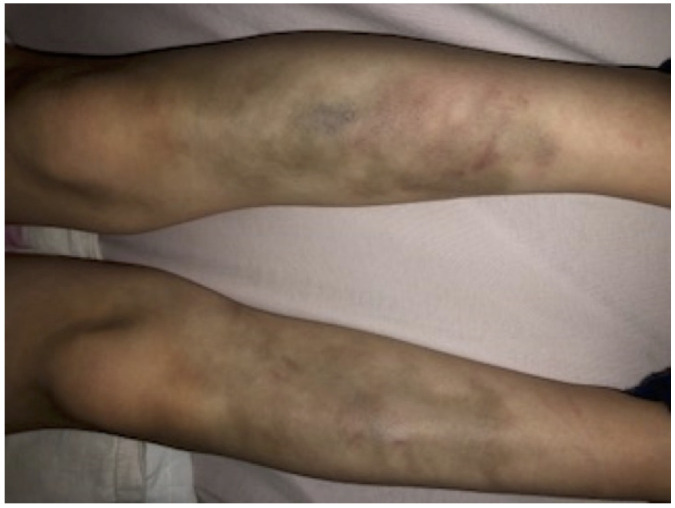
Pretibial findings. Easy bruising is a common clinical manifestation of both classical and periodontal EDS and can occur anywhere on the body, including unusual sites like the cheeks. The pretibial area often remains stained with hemosiderin from previous bruises.

**Figure 4 biomolecules-11-00149-f004:**
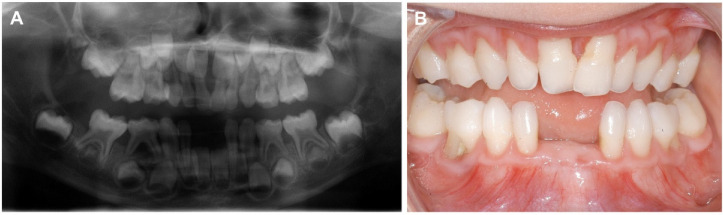
Radiological and clinical dental investigation. (**A**) The panoramic X-ray at age five years revealed extensive alveolar bone loss on teeth 82, 83, and 84 with furcation involvement in the latter. Teeth 71 and 81 were lost due to periodontal bone loss at age four years. (**B**) Lack of attached gingiva is evident in the mandibula; the thin and fragile mucosa extends to the free gingival margin and the interdental papillae, which are normally keratinized and “thick”. Severe gingival recession up to 6 mm was diagnosed in all teeth but numbers 75 and 85.

**Table 1 biomolecules-11-00149-t001:** Clinical Features of classical and periodontal Ehlers-Danlos syndrome (EDS) in the present family.

	Proband IV:2	Mother III:2	Grandfather II:2
**Major criteria of classical EDS**			
Joint hypermobility	+	+	+
Hyperelastic and soft skin	+	+	−
Atrophic scars	+	+	n.a.
**Major criteria of periodontal EDS**			
Early severe periodontitis	+	−	−
Lack of attached gingiva	+	−	n.a.
Pretibial hyperpigmentation	+	−	−
**Minor criteria of classical and / or periodontal EDS**			
Acrogeric face and hands	−	−	−
Myopia	−	+	−
Blue sclerae	+	−	
Ectopia lentis	−	−	−
epicanthic folds	−	+	
Arachnodactyly	−	−	−
multiple subcutaneous papules	+	+	
Easy bruising	+	+	+
Poor wound healing	+	+	+
dilatation of aorta	−	−	n.a.
Hernias	−	umbilical hernia	−
Organ rupture	−	−	−
Recurrent infections	+	+	+
Involvement of heart valves	hypermobile mitral valve (no prolapse)	n.a.	n.a.
Luxation of knee caps	+	+	

## Data Availability

The datasets supporting the conclusions of this article are available at the Department of Cranio-Maxillofacial Surgery, University Hospital Münster, Germany.

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
