# Peer review of "Prepubertal Periodontitis in a Patient with Combined Classical and Periodontal Ehlers–Danlos Syndrome"

_biomolecules, 2021, doi:10.3390/biom11020149_

Round 1

Reviewer 1 Report

In this report, the authors describe the clinical features of a family of German ancestry, presenting classical type of Ehlers-Danlos Syndrome, caused by a pathogenic variant in COL5A1, and in particular the phenotype of one family-member, a nearly 5-year old girl who in addition presented loss of two incisors, alongside severe periodontal destruction and lack of attached gingiva. This prompted the authors to perform additional molecular screening, revealing in this girl a second defect, i.e. a novel likely pathogenic variant in C1R. this confirms a dual diagnosis of both periodontal and classical EDS in this girl. 

The authors underscore the fact that one should be open-minded regarding the possibility of two monogenic disorders in one patient, and that early severe periodontal destruction with lack of attached gingiva is a specific finding of periodontal EDS but not of other EDS types.

The case report is very well written and clearly illustrated. Introduction and discussion are to the point, and well-referenced. I have no major remarks. 

There are a few typographical errors; here are the ones that I detected: 

  • Page 3-line 119: radiological
  • Page 6- line 152: massive
  • Page 6- line 163: C1S --> italic font
  • Page 7-line 199: distinguish
  • Page 7-line 204: especially

Author Response

We would like to thank the editor and the reviewers for their time spent on reviewing our manuscript and their helpful comments. Their suggestions have been implemented in the manuscript. In this letter, we respond point-by-point to the comments and explain the revisions.

All changes to the manuscript were highlighted using the "Track Changes" function in Microsoft Word.

We hope the manuscript is now suitable for publication in Biomolecules.

Reviewer 1:

In this report, the authors describe the clinical features of a family of German ancestry, presenting classical type of Ehlers-Danlos Syndrome, caused by a pathogenic variant in COL5A1, and in particular the phenotype of one family-member, a nearly 5-year old girl who in addition presented loss of two incisors, alongside severe periodontal destruction and lack of attached gingiva. This prompted the authors to perform additional molecular screening, revealing in this girl a second defect, i.e. a novel likely pathogenic variant in C1R. this confirms a dual diagnosis of both periodontal and classical EDS in this girl. 

The authors underscore the fact that one should be open-minded regarding the possibility of two monogenic disorders in one patient, and that early severe periodontal destruction with lack of attached gingiva is a specific finding of periodontal EDS but not of other EDS types.

The case report is very well written and clearly illustrated. Introduction and discussion are to the point, and well-referenced. I have no major remarks. 

There are a few typographical errors; here are the ones that I detected: 

  • Page 3-line 119: radiological
  • Page 6- line 152: massive
  • Page 6- line 163: C1S --> italic font
  • Page 7-line 206: distinguish
  • Page 7-line 204: especially

Answer: We corrected the typographical errors page 3 line 131, page 6 line 164 and 175, page 7 line 211 and 217.

Reviewer 2:

In the manuscript entitled: “Prepubertal periodontitis in a patient with combined 2 classical and periodontal Ehlers-Danlos syndrome” the authors investigated rare case of combined classical and periodontal Ehlers−Danlos syndrome (EDS) with early severe periodontitis and a generalized lack of attached gingiva.

The authors concluded that due to these clinical findings an additional diagnosis of periodontal EDS was suspected. A further genetic analysis revealed the novel missense mutation c.658T>G (p.Cys220Gly) in C1R in a heterozygous state. Early severe periodontitis in association with generalized lack of attached gingiva is pathognomonic for periodontal EDS and led to the right clinical and genetic diagnosis in the present case.

Major comments:

In general, the idea and innovation of this study, regards the analysis of periodontitis in Ehlers-Danlos syndrome patients is interesting, because the role of periodontitis and systemic diseases released during wound healing are validated, but further studies on this topic could be an innovative issue in this field could be open a creative matter of debate in literature by adding new information. Moreover, there are few reports in the literature that studied this exciting topic with this kind of study design. The study was well conducted by the authors; However, there are some concerns to revise that are described below.

Comment 1: “The introduction section and the manuscript resume the existing knowledge regarding the important factor linked with mediators released during inflammation. However, as the importance of the topic, the reviewer strongly recommends, before a further re-evaluation of the manuscript, to update the literature through read, by must discuss and cites in the references with great attention all of those recent interesting articles, that helps the authors to better introduce and discuss the aim of the study in light of others mediators released following periodontitis: Isola G, Lo Giudice A, Polizzi A, Alibrandi A, Murabito P, Indelicato F. Identification of the different salivary Interleukin-6 profiles in patients with periodontitis: A cross-sectional study. Arch Oral Biol. 2020 Nov 30;122:104997. doi: 10.1016/j.archoralbio.2020.104997. 2); Isola G, Polizzi A, Alibrandi A, Williams RC, Leonardi R. Independent impact of periodontitis and cardiovascular disease on elevated soluble urokinase-type plasminogen activator receptor (suPAR) levels. J Periodontol. 2020 Oct 22. doi: 10.1002/JPER.20-0242. 3); Isola G, Polizzi A, Patini R, Ferlito S, Alibrandi A, Palazzo G. Association among serum and salivary A. actinomycetemcomitans specific immunoglobulin antibodies and periodontitis. BMC Oral Health. 2020 Oct 15;20(1):283. doi: 10.1186/s12903-020-01258-5.”

Answer: We added a paragraph on the pathogenesis of periodontitis to the introduction. We also included references which the reviewer recommended. As periodontal destruction with pEDS is caused by an overactivation of complement 1, we mainly focused on the classical complement pathway but not on other pathways.

Comment 2: “The authors should be better specified at the end of the introduction section, the rationale of the study.”

Answer: We stated now more precisely the rationale of our case report at the end of the introduction section (line 89-97).

Comment 3: “In the discussion, should better clarify the role of periodontal inflammation in the risk development of EDS.”

Answer: We added a paragraph on inflammatory conditions and their association with EDS (line 222-226).

Comment 4: “The conclusion should be added with the main findings of the study and reinforce in light of the future directions.”

Answer: We added a conclusion section with the main findings and reinforced the need of future cohort studies on periodontal and dental manifestations with EDS (line 229-234).

In conclusion, I am sure that the authors are excellent clinicians who achieve very nice results with their adopted protocol. However, this study, in my view, does not in its current form satisfy a very high scientific requirement for publication in this journal and requests a revision before a further re-evaluation of the manuscript.

Minor Comments:

Introduction:

Comment 5: “Please refer to major comments”

Answer: We hope that we could satisfactorily address the major comments (see above)

Discussion

Comment 6: “Please add a specific sentence that clarifies the results obtained in the first part of the discussion.”

Answer: We have now added a sentence that we hope clarifies the findings of periodontal destruction in the primary dentition (line 198-200).

Reviewer 2 Report

In the manuscript entitled: “Prepubertal periodontitis in a patient with combined 2 classical and periodontal Ehlers-Danlos syndrome” the authors investigated rare case of combined classical and periodontal Ehlers−Danlos syndrome (EDS) with early severe periodontitis and a generalized lack of attached gingiva.

The authors concluded that due to these clinical findings an additional diagnosis of periodontal EDS was suspected. A further genetic analysis revealed the novel missense mutation c.658T>G (p.Cys220Gly) in C1R in a heterozygous state. Early severe periodontitis in association with generalized lack of attached gingiva is pathognomonic for periodontal EDS and led to the right clinical and genetic diagnosis in the present case.

Major comments:

In general, the idea and innovation of this study, regards the analysis of periodontitis in Ehlers-Danlos syndrome patients is interesting, because the role of periodontitis and systemic diseases released during wound healing are validated, but further studies on this topic could be an innovative issue in this field could be open a creative matter of debate in literature by adding new information. Moreover, there are few reports in the literature that studied this exciting topic with this kind of study design.

The study was well conducted by the authors; However, there are some concerns to revise that are described below.

The introduction section and the manuscript resume the existing knowledge regarding the important factor linked with mediators released during inflammation.

However, as the importance of the topic, the reviewer strongly recommends, before a further re-evaluation of the manuscript, to update the literature through read, by must discuss and cites in the references with great attention all of those recent interesting articles, that helps the authors to better introduce and discuss the aim of the study in light of others mediators released following periodontitis: 1) Isola G, Lo Giudice A, Polizzi A, Alibrandi A, Murabito P, Indelicato F. Identification of the different salivary Interleukin-6 profiles in patients with periodontitis: A cross-sectional study. Arch Oral Biol. 2020 Nov 30;122:104997. doi: 10.1016/j.archoralbio.2020.104997. 2) Isola G, Polizzi A, Alibrandi A, Williams RC, Leonardi R. Independent impact of periodontitis and cardiovascular disease on elevated soluble urokinase-type plasminogen activator receptor (suPAR) levels. J Periodontol. 2020 Oct 22. doi: 10.1002/JPER.20-0242. 3) Isola G, Polizzi A, Patini R, Ferlito S, Alibrandi A, Palazzo G. Association among serum and salivary A. actinomycetemcomitans specific immunoglobulin antibodies and periodontitis. BMC Oral Health. 2020 Oct 15;20(1):283. doi: 10.1186/s12903-020-01258-5.

The authors should be better specified at the end of the introduction section, the rationale of the study. In the discussion, should better clarify the role of periodontal inflammation in the risk development of EDS.

The conclusion should be added with the main findings of the study and reinforce in light of the future directions.

In conclusion, I am sure that the authors are excellent clinicians who achieve very nice results with their adopted protocol. However, this study, in my view, does not in its current form satisfy a very high scientific requirement for publication in this journal and requests a revision before a further re-evaluation of the manuscript.

Minor Comments:

Introduction:

  • Please refer to major comments

Discussion

  • Please add a specific sentence that clarifies the results obtained in the first part of the discussion

Author Response

(The authors gave the same response as above.)

Round 2

Reviewer 2 Report

The authors have well addressed to all reviewers comments. I suggest the acceptance of this interesting manuscript.